# Unravelling Vaccine Scepticism in South Tyrol, Italy: A Qualitative Analysis of Personal, Relational, and Structural Factors Influencing Vaccination Decisions

**DOI:** 10.3390/healthcare11131908

**Published:** 2023-07-01

**Authors:** Christian J. Wiedermann, Peter Koler, Sara Tauber, Barbara Plagg, Vera Psaier, Verena Barbieri, Giuliano Piccoliori, Adolf Engl

**Affiliations:** 1Institute of General Practice and Public Health, Claudiana College of Health Professions, 39100 Bolzano, BZ, Italy; 2Department of Public Health, Medical Decision Making and Health Technology Assessment, University of Health Sciences, Medical Informatics and Technology, 6060 Hall, Austria; 3Nonprofit Organization Forum Prevention, 39100 Bolzano, BZ, Italy; 4Faculty of Education, Free University of Bolzano, 39100 Bolzano, BZ, Italy

**Keywords:** vaccination, vaccine refusal, social control, informal, health knowledge, attitudes, practice, complementary therapies, health knowledge, attitudes, practice

## Abstract

Low vaccine uptake in South Tyrol, particularly for non-coronavirus and SARS-CoV-2 vaccines, poses a significant public health challenge in the northernmost province of Italy. This qualitative study conducted in-depth interviews with a purposive sample of vaccine-sceptical parents to examine the factors that contribute to their vaccination decisions. The ten participants’ children had varied vaccination statuses, ranging from unvaccinated to partially vaccinated or vaccinated as late as possible. Only one adult participant received the SARS-CoV-2 vaccine. Using Grounded Theory analysis, the emergent meta-category of ‘self-relatedness’ was identified, highlighting the importance of individual experiences and the social context. The study found that participants’ social circles consisted of individuals with similar vaccination attitudes, often characterized by a shared affinity for nature. Although they accepted individuals with different views, they remained uninfluenced. Participants perceived healthcare professionals as one-sided and uncritical, expressing distrust toward state orders. They believed that parents should be responsible for their children’s well-being rather than the state. Distrust in the state and healthcare system, exacerbated by the SARS-CoV-2 pandemic, was rooted in negative experiences. In contrast, the participants had positive experiences with natural healing, homeopathy, and trusting the natural course of events. They perceived themselves as tolerant, non-radical, curious, health-conscious, yet critical and questioning. Participants resisted coercion, fear-mongering, and state sanctions and sought alternatives to mandatory vaccination. To address the complex social and behavioural factors underlying vaccination refusal, this study suggests that vaccination advocates, policymakers, and information providers should engage in appreciative, personal, and well-founded information exchanges with vaccine-hesitant individuals. Broad and comprehensible information dissemination, flexibility, and freedom of decision are essential for increasing informed decision making. Further research is required to better understand the epistemic basis of vaccine hesitancy.

## 1. Introduction

Preventing diseases through vaccination is one of the greatest achievements in medicine and has been successfully demonstrated during the coronavirus pandemic. High rates of vaccine hesitancy have been reported globally during the coronavirus disease 2019 (COVID-19) pandemic, forming a significant barrier to vaccine uptake [1]. In high-income countries, rates of vaccine hesitancy have ranged from 7% to 77.9%, with nearly half of the studies reporting hesitancy rates of 30% or more. Various factors contribute to this hesitancy, including individual and group factors such as a lack of recent influenza vaccination history, lower perceived risk of contracting COVID-19, and beliefs about the safety and efficacy of vaccines. Significant efforts have been made to reduce hesitant vaccination behaviours and to increase vaccine uptake. Social norms and individual psychological characteristics are among the many determining factors behind vaccine hesitancy [2] and are important for understanding vaccination decisions [3].

Socioeconomic and demographic factors that influence community resilience determine social vulnerability [4]. Social vulnerability, defined as the susceptibility of social groups to potential harm due to factors like socioeconomic disadvantages or limited resource access, can influence vaccine acceptance. This vulnerability can breed mistrust in health systems, including vaccinations, and lead to vaccine hesitancy. On the other hand, lower psychological resilience predicts willingness to take the vaccine when offered [5]. People living in areas with higher social vulnerability are more likely to hesitate to report SARS-CoV-2 vaccine hesitancy [6].

Vaccine scepticism and hesitancy differ across countries because of variations in cultural, political, social, and economic factors. Although there may be some commonalities in the factors influencing vaccination decisions, the relative importance of each factor and specific concerns raised by individuals may vary.

For instance, in some countries, trust in the government and healthcare institutions might be a more significant factor in vaccine hesitancy, whereas in others, cultural or religious beliefs may play a more prominent role. In addition, the history of vaccination policies and public health campaigns in each country can influence the public’s perception of vaccines.

Representative information on vaccination intention is important for an agile regional healthcare response to vaccination [7]. By conducting comparative research across countries, researchers can identify common themes and differences in factors that influence vaccination decisions. This information can then be used to develop tailored strategies to address vaccine scepticism and hesitancy.

In Italy, mandatory vaccination for newborns and children is regulated by a decree called “Legge n. 119/2017”, also known as “Lorenzin Law”, named after the then Minister of Health, Beatrice Lorenzin. This law mandates ten vaccinations for children aged 0 to 16 years, including polio, diphtheria, tetanus, hepatitis B, pertussis, Haemophilus influenzae type b, measles, mumps, rubella, and varicella. Parents who refuse to vaccinate their children can face fines ranging from EUR 100 to EUR 500, depending on the regional regulations. Non-vaccinated children may also be denied access to educational institutions, including nursery schools and kindergartens. However, compulsory vaccination is not a requirement for attending primary schools and above, but parents are still subject to fines [8]. Research conducted among public health workers in Italy has shown significant lifestyle changes during the COVID-19 period, such as increased screen time and changes in diet, sleep patterns, and physical activity, all indicators of heightened stress and workload [9]. This context underlines the critical importance of studying the drivers behind vaccine acceptance and hesitancy, as health workers’ attitudes towards vaccines can significantly influence the wider population’s uptake and overall pandemic management.

In Italy, SARS-CoV-2 and mandatory non-coronavirus vaccination rates are among the lowest in South Tyrol, the northernmost province, to which vaccine hesitancy and resistance could be contributing factors [10]. South Tyrol, located in northern Italy with a catchment area of 530,000 inhabitants, boasts a unique demographic makeup. It is an autonomous province with a largely German-speaking population (about 70%), while the rest are mainly Italian and Ladin speakers. The region has a strong cultural identity that blends Alpine and Mediterranean influences, with distinct beliefs, traditions, and healthcare practices. Historically, South Tyrol has been marked by political and cultural tensions, which may impact trust in public health authorities and initiatives, thereby influencing vaccine hesitancy. Moreover, the region’s geographic isolation and strong reliance on local community networks could play a role in the spread and entrenchment of vaccine-sceptical attitudes. Therefore, an understanding of vaccine hesitancy in South Tyrol may offer valuable insights into how local socio-cultural factors influence vaccine-related beliefs and behaviours.

Qualitative methods are beneficial for exploring vaccine hesitancy due to their ability to capture the nuanced interplay of perceptions, beliefs, and socio-cultural contexts that influence individuals’ decisions. They provide an in-depth understanding of personal experiences and local contexts, which is critical for devising effective strategies to address vaccine hesitancy in specific communities. Internationally, qualitative research has delved into the underlying factors influencing vaccine hesitancy in high-income countries [11] and the social cultivation of vaccine refusal in specific community settings, like Waldorf schools [12]. Previous qualitative studies in Italy have shed light on vaccine hesitancy, with research revealing the significant role of information systems in countering such hesitancy [13] and the manifestation of vaccine hesitancy among different population groups, such as university students [14]. These studies offer valuable insights into the cultural, socioeconomic, and individual factors shaping vaccine hesitancy, informing our approach to the current investigation in Northern Italy.

Our study aims to explore the relationship between social vulnerability and vaccine scepticism in Northern Italy to inform public health. Because of the complexity of the investigated behavioural and social phenomena and interactions, we used a qualitative approach in the present study to further investigate vaccination intentions in that particular region.

## 2. Methods

### 2.1. Sample

The interview participants were recruited by the Institute of General Medicine and Public Health at the Claudiana College of Health Professions, Bolzano (BZ). The inclusion criteria for the interviewees were determined based on an ongoing examination of the data of a quantitative study [10]. However, the prerequisite for participation was the personal assessment of potential interview partners, such as vaccination sceptics or opponents, as well as being parents.

Regarding the sample, the importance of including individuals with relevant sample characteristics emerged during the research process. Where possible, the sample acquisition was adapted to the initial results of the research question.

Some ambivalence was observed regarding participation in the study. A few individuals who showed principled willingness to participate could be located. People repeatedly withdrew their wishes to participate after their initial unofficial commitment. Participation in the interviews was voluntary, the data were processed anonymously, and the participants provided written consent to participate, including permission to further process the acquired data.

Ten interviews were conducted in total. Seven interview participants were female (70%) and three were male (30%). On average, the persons were 42.5 (range 37–52) years old. All interview participants had children. On average, the interviewees had 2.5 children (range = 1–5); they were between 3 months and 21 years old at the time of the interview. Two (20%) interviews were conducted in Italian and eight (80%) in German. Eight (80%) interview participants were assigned to a rural area and two (20%) to an urban area of South Tyrol. Six of the interview participants had a university degree (60%), three attended secondary school (30%), and one completed an apprenticeship (10%).

### 2.2. Questionnaire

To define the research question, various discussions took place between the staff of the involved institutions, Forum Prevention and the Institute of General Practice and Public Health, in Bolzano (BZ), Italy. Theoretical presuppositions and content from a quantitative study of vaccination attitudes regarding SARS-CoV-2 and compulsory non-coronavirus vaccinations in South Tyrol [10] were discussed, and topics to be deepened in the interviews were jointly identified.

The following specific research question emerged from this initial analysis process: “What factors play a role in the relatively low vaccination readiness in South Tyrol?” Based on this, the following main topics and objectives emerged for this research project:Recording of attitudes as well as behaviours on the topic of vaccination;Survey of the interests and needs of people who are sceptical about vaccination/refusal of compulsory vaccinations;Survey of the causes and reasons for vaccination scepticism/refusal to vaccinate;Survey of the effects of current framework conditions on persons who are sceptical about the topic of vaccination/refusal of compulsory vaccinations.

Since no sufficient findings were expected from the described quantitative study for the social and behavioural research questions and the above topic areas, an inductive approach was proposed. Thus, the grounded theory approach appears to be suitable. Semi-structured interviews were conducted using a semi-standardized questionnaire. For the semi-structured interviews, the research group developed an interview guide with open-ended questions based on the macro areas identified within the focus group.

#### Question Guide

After preliminary interviews, an initial question guide was created. The relevance of the questions and their validity in relation to the research question were evaluated in a pretest. The subsequent examination of the first generated data material led to changes in the questions, as well as to a deepening of some topics. During the research process, new questions and content that needed to be explored in greater depth were continuously incorporated into the questionnaires.

The questions were adapted to current conditions in relation to the topic of vac-cination. In particular, regarding SARS-CoV-2 vaccination, certain environmental conditions changed during the research process, and these were considered in the in-terviews and, thus, in the question guide.

The questionnaire contained information on the implementation, research ques-tion, and topics of the research project; the request for socio-demographic data col-lected prior to the interviews; and questions on the central topics described above. It was translated into German and Italian. Further personal data of the interviewees, re-sulting from the interviews, were anonymized during the course of the transcription.

### 2.3. Data Collection and Analysis

Data collection began in April 2021 and was completed in October 2021. The interviews were carried out by three trained researchers. Prior to the interview process, all three researchers received specific briefings regarding the topic of vaccination to ensure consistency and depth in their queries. These researchers were actively involved in the entire research process.

Principles of interviewing were as follows:Impartiality with regard to the subject matter;Open and non-judgmental conduct of the interview;Validating basic attitude for the attitudes and behaviours of the interviewees.

The interviews were conducted in person at the interviewees’ homes or at the headquarters of Forum Prevention in Bolzano (BZ). All interviews were audio-recorded, each comprising a total duration of approximately 38 min to 1.5 h. The recorded audio files of the interviews were deleted from all data carriers after transcription.

The interviews were transcribed. The data were analysed using qualitative content analysis [15]. Accordingly, the interviews were coded, grouped into categories, and analysed. Each transcript was coded by two researchers to ensure reliability, and further interpretation and discussion were conducted within the entire research team to ensure that the analytical deductions were congruent with the extracts. The interpretation of data in this study was an iterative process following the Grounded Theory approach. Initial open coding of the interviews was conducted, identifying primary themes within the data. These initial codes were then further grouped into conceptual categories in a process known as axial coding, ensuring that our findings were firmly rooted in our participants’ responses. Constant comparison was applied, wherein each new piece of data was compared with existing codes and categories, to ensure the emergent themes accurately reflected the data. Interpretation of these emergent themes was thus directly derived from, and validated by, our interview data.

#### 2.3.1. Grounded Theory

Grounded Theory, developed by Glaser and Strauss [16], is a research methodology that aims to construct theories based on data collected from social reality. This approach is characterized by its adaptability to the research subject and compatibility with other methods, including quantitative data collection. The theory-building process applied in this study, as outlined by Strauss et al. [17], consists of several steps: identifying concepts for phenomena (analytical categories), relating these categories to each other, dimensionalizing them, and ultimately integrating them into a coherent theory [18]. Grounded Theory ensures that the resulting theories are firmly rooted in empirical data and are relevant to the social context being studied.

#### 2.3.2. Coding

The interviews were analysed by the three Forum Prevention staff members involved in the research process. Specifically, the first interviews were coded to create a preliminary category system. During the research process, this was repeatedly checked for validity and adapted to the interview results. This led to the emergence of superordinate categories, and the connections between categories became clear. The categories were discussed, and their contents were defined and clarified using examples until the individual superordinate categories and subcategories exhibited a sufficient degree of selectivity in comparison.

### 2.4. Criteria for Reporting

We have adhered to the Consolidated Criteria for Reporting Qualitative Research (COREQ) [19] to ensure rigorous and transparent reporting of our study. The completed COREQ checklist is provided as a Appendix A to this manuscript, detailing the research team, study design, analysis, and findings to promote a comprehensive understanding and evaluation of our qualitative approach.

## 3. Results

### 3.1. Meta-Category

The emergent meta-category identified by the analysis of the data using Grounded Theory in this study is “self-relatedness”, which distinguishes itself by being able to relate to underlying categories and create meaningful connections among them. Self-relatedness should not be equated with selfishness; instead, it indicates that participants placed a strong emphasis on understanding how environmental factors impact themselves and their families. Participants recognized the importance of societal and individual health, but seldom viewed personal behavioural changes as a means of promoting overall public health.

The lack or weak presence of trust in the health system further contributed to this stance. Information from close social circles, self-acquired knowledge, and interpersonal exchanges hold more persuasive power than the information disseminated by the health system. The importance of social groups in shaping opinions is also evident, as individuals often seek confirmation from like-minded groups.

Building upon self-relatedness, a fundamental attitude emerges that is driven by selectively perceived experiences, encounters, and events, leading to the development of convictions. Labelling participants as “No-vax” generates resistance, as they do not identify with this social phenomenon and perceive it as a media creation. This label contributes to the solidification of vaccine scepticism, moving from latent ambivalence to conviction. Participants displayed high sensitivity to health issues, engaged in both conventional and alternative treatments, and preferred making informed decisions based on their understanding of health and illness.

### 3.2. Key Moments

The category “Key moments” (44 codes) consists of two subcategories: “First confrontation” (26 codes) and “Activation moments” (18 codes). This category dealt with situations in the lives of the interviewees that were crucial for the salience and importance of the vaccination issue, as well as the moments and experiences that led to initial scepticism about vaccinations. The first confrontation with the topic of vaccination often occurs when individuals become parents, prompting them to seek information.


*“Since I had my first child.”*

*(A06: 17–18)*


This statement highlights the key moment in becoming a parent as the first confrontation with vaccination. This reflects the common experience of individuals seeking information and developing an interest in the vaccination issue upon becoming parents. This moment often triggers the process of questioning and scepticism regarding vaccination. Other situations that led to vaccination scepticism included workplace recommendations, upcoming travel, and the observation of newborn vaccination.

Activation moments, however, involve incidents or situations in which individuals develop doubts about vaccination beyond just seeking information. These moments included instances where potential side effects or the potential harm of vaccinations became apparent as well as experiences of allergic reactions or health issues related to vaccinations. Activating moments are sometimes intertwined with the first confrontation, such as after the birth of a child, where feelings of responsibility and vulnerability lead to information seeking and, ultimately, doubt. Additionally, experiences with the healthcare system or the compulsion to vaccinate also served as activation moments for some interviewees.

### 3.3. Vaccination Verities

#### 3.3.1. Vaccination-Related Information

This study identified two main categories of vaccination-related information: mandatory vaccinations and COVID-19 vaccinations. The first category included subcategories such as sources of information, knowledge of mandatory vaccinations, scepticism about vaccinations, and attitudes toward mandatory vaccinations. The second category included subcategories such as attitudes toward COVID-19, sources of information and knowledge about COVID-19 vaccinations, scepticism about COVID-19 vaccinations, and attitudes toward COVID-19 vaccinations.


*“We have read various books, both pro-vaccination and contra-vaccination. At some point, we stopped because, as I said, you can find the exact opposite opinion on everything. However, obtaining a clear view is difficult. It’s not just me or us, but we have experienced this with many others as well.”*

*(A09: 96)*


This statement reflects the variety of information sources that parents use to learn about vaccination, including books and conversations with others. It also highlights the difficulty of finding consistent information and forming a clear opinion owing to the diverse and sometimes contradictory viewpoints available.

The results showed that parents used a variety of information sources, including social networks, medical professionals, alternative medicine practitioners, books, websites, and the media.

#### 3.3.2. Vaccination Knowledge

The findings suggest that knowledge related to mandatory vaccinations is mostly negative and includes aspects that can raise doubts about the effectiveness and safety of vaccinations, which could be considered important for vaccine sceptics. The subjective knowledge collected included information about the number of mandatory vaccinations in Italy compared with other European countries, ingredients in vaccines, administration timing and modality, necessity of certain vaccinations, risks associated with vaccinations, and history of vaccination. Additionally, vaccine sceptics tend to have a more comprehensive and varied information spectrum on the topic than vaccine supporters. Overall, the study highlights the need for better communication and education regarding mandatory vaccinations to address misinformation and misunderstanding.

#### 3.3.3. Vaccination Scepticism

The study identified five major themes that contributed to vaccination scepticism: Hepatitis-B vaccination, the number of mandatory vaccinations in Italy compared to other countries, the economic interests of the pharmaceutical industry, knowledge of vaccine injuries, and the handling of mandatory vaccinations by legislation. They also expressed scepticism regarding the necessity of mandatory vaccinations against certain diseases and the motives of the pharmaceutical industry to produce them. Knowledge of vaccine injuries and personal experiences with the adverse effects of vaccination also contributed to scepticism. Participants were also critical of the handling of mandatory vaccinations by legislation, including the inconsistency of mandatory vaccinations, ease of obtaining exemptions, and the perceived lack of a comprehensive assessment of vaccine risks by the medical community.


*“I wanted to understand why a two-month-old baby was given a vaccine for hepatitis, which is a sexually transmitted disease. I wanted to understand why vaccines are administered for diseases that are not lethal, so to speak, and are treatable. This is particularly true after the first few years of life. A disease like whooping cough at three years of age is no longer dangerous and treatable.”*

*(A06: 36)*


This statement reflects scepticism and concerns about the necessity of mandatory vaccinations against certain diseases, the timing of their administration, and the potential risks associated with vaccination.

Overall, the findings suggest that vaccine scepticism is driven by a combination of factors, including distrust of the pharmaceutical industry and government, perceived inconsistency in vaccination policies, and concerns regarding the potential risks associated with vaccination.

#### 3.3.4. Attitudes toward Mandatory Vaccination and COVID-19

Findings on attitudes toward mandatory vaccination and COVID-19 describe the personal attitudes of interviewees toward vaccinations, including their views on COVID-19 and the measures taken in response to the pandemic. Interviewees were asked to rate their attitudes toward vaccination on a scale of one to ten. Although all participants described themselves as vaccine sceptics, their attitudes toward vaccinations varied widely. Some participants stated that they would never vaccinate themselves or their children, whereas others were open to vaccination under certain circumstances. The majority of interviewees did not consider COVID-19 a significant threat and believed that the measures taken to prevent its spread were excessive. They also criticized the media for stirring up fear and expressed concerns about the impact of COVID-19 measures on mental health and employment. Some interviewees were critical of masks, arguing that they were ineffective or potentially harmful.


*“First of all, I would like to say that I am neither a vaccination advocate nor an opponent; I believe that everyone should weigh the pros and cons for themselves.”*

*(A08: 2)*


This statement best reflects the range of personal attitudes toward vaccinations described in the text, acknowledging the individual nature of decision making without taking an extreme stance in either direction.

Overall, interviewees stressed the importance of personal choices and responsibility in decision making regarding vaccination and COVID-19 measures.

#### 3.3.5. Information Sources and Knowledge on COVID-19 Vaccination

This study found that it was difficult to separate the information sources for vaccination from those for other COVID-19 containment measures. Participants received information from various sources, including their social circles (friends, family, and some with medical backgrounds), the Internet, foreign newspapers, social media posts and videos, documentaries on TV, and the national vaccination campaign. Some participants obtained information through research, whereas others received it incidentally.

Participants also mentioned that they limit themselves to gathering information as the topic is overwhelming, and the more they research, the more their scepticism and dissatisfaction increase. They also noted that traditional media (newspapers, television) are not used much, and are sometimes viewed sceptically because they tend to present only the pro-vaccination and pro-measures views, while social media is perceived as a place of polarization. Some participants mentioned that traditional media are not reliable and do not show the whole truth.

Regarding COVID-19 vaccination information sources, some participants reported that they obtained information from a group of over one hundred doctors who had written open letters expressing concerns about the vaccine. Others mentioned that they obtained information about COVID-19 measures such as lockdowns from sources such as the WHO, an independent German investigation committee, and an alternative exchange congress.

The study also found that participants had various knowledge and information related to the COVID-19 vaccine, such as recommendations for the administration of the vaccine, the need to refresh the vaccine every six months, the different types of vaccines, their risks and side effects, and the communication strategies used to promote vaccination.


*“Yes, where do I get my information? It’s more by chance. I have also researched a bit.”*

*(A02: 48)*



*“Of course, because we were constantly exposed to it. If you listen to news or something like that, it is always a topic. Therefore, it was more accessible, much more intense, and—I must say, mainly the pro-line, at least on public channels. What happens on social media is, of course, the usual polarization. Pro-opinion is predominant in traditional mass media. [...]”*

*(A09: 130)*


These statements reflect the various information sources participants mentioned and how they perceived traditional and social media, as described.

#### 3.3.6. Concerns, Doubts, and Personal Autonomy in COVID-19 Vaccine Scepticism

Various factors contributed to scepticism toward COVID-19 vaccination among the individuals interviewed. These factors include concerns about the speed of vaccine development, the novelty of the vaccine, and the potential for vaccine damage years after vaccination. Scepticism is also reinforced by statements from medical professionals expressing doubt and suspicion about the profit motives of pharmaceutical companies and organizations, such as the WHO.


*“And I am even more sceptical about this than about other vaccinations because it is completely new and has nothing to do with traditional vaccinations. It is completely different and has been brought to the market and approved in such a short time. The number of tests performed was relatively small. [...]”*

*(A08: 64)*


This statement best reflects concerns regarding the speed of vaccine development, the novelty of the vaccine, and potential side effects.

In general, interviewees emphasized their doubts about cost–benefit analysis in relation to personal health. The possible side effects, regardless of the specific vaccine, outweigh the perceived benefits of COVID-19 vaccination for most people (especially young people and children). Interviewees described the perception that there is pressure to get vaccinated and that measures are aimed at encouraging people to get vaccinated, even if they are unsure about it. Compulsory vaccination is condemned, particularly for younger people, as it is not seen as health-promoting and should not infringe on personal freedom.

The polarization of society and the resulting conflicts arising from the issue were also rejected by some interviewees. In general, it is emphasized that everyone should make their own decisions about whether to be vaccinated.

These findings suggest that, for the most part, vaccine sceptics will not be vaccinated against COVID-19. However, some exceptions exist, with one interviewee stating that they would be vaccinated to protect the public, even though they are personally not convinced that vaccination is the only way out.

### 3.4. Vaccination Reality

The “Vaccination Reality” category contained 15 subcategories with 326 codes. This category collects coded interview statements that contribute significantly to developing one’s attitude toward the vaccination of their own children and the COVID-19 vaccine. It shows the clash between personal views and truths about vaccination with “reality”. The categories go beyond assumptions and information to show the factors that ultimately led to vaccinating or not vaccinating and the related consequences. These categories help us understand the factors that influence individual decisions and what justifies and explains these decisions. This includes influencing factors (63 codes) and reasons for the decision to vaccinate (31 codes) or not vaccinate (57 codes).

There are also separate subchapters on COVID-19 (codes 13, 1, 15). Since the interviewees were vaccine sceptics and opponents, the decision not to vaccinate outweighed the decision to vaccinate. Therefore, some of the subchapters deal with the consequences of vaccination (13 codes plus two codes for COVID-19 vaccination) and not vaccinating (46 codes), as well as not vaccinating for COVID-19 (13 codes), dealing with vaccination mandates (22 codes), the act of vaccination (9 codes), and possible motivation for change (15 codes). Two additional chapters summarize the codes for the ambivalence (19 codes) and cost-benefit assessment (7 codes) categories.

#### 3.4.1. Influencing Factors for Vaccination Decision

The subcategory collects statements about factors that significantly influence vaccination decisions, primarily the decision not to vaccinate one’s own child.

According to the interviewees, the acquisition of information—self-education and questioning received—appears to be an important factor. The more knowledge the interviewees acquired on the topic, the more sceptical they seemed to be about vaccination. The topics of health and the possibility of health risks for the child due to knowledge of vaccine damage are often mentioned. Vaccine damage becomes more significant if such cases occur in families or in social circles. The timing of vaccination when children are too small at the time of vaccination, is often mentioned. The number of vaccinations administered simultaneously against multiple diseases has created scepticism.

Having no choice to individually weigh the pros and cons of each vaccination and deciding for each individual is described as a disturbing element. The ability to make decisions is a key factor. This competence is experienced as a valuable part of one’s self, which they are unwilling to abandon. Decision making freedom is essential, and if it is removed, the probability of deciding against what is imposed increases.


*“However, when we delved deeper into this whole story, she also came to the conclusion that it was not the right decision for us to vaccinate our children.”*

*(A08: 26)*


This statement highlights the process of self-education and questioning received information, which leads to the decision to not vaccinate their children. It also touches upon scepticism toward vaccination, which is a central theme in the text.

However, some topics make the decision to vaccinate easier, such as desire to return to work, external pressure, financial needs, and childcare needs. A cost–benefit analysis regarding a child’s overall well-being may also lead to the conclusion that the decision to vaccinate is better, especially when it comes to friendship contacts for the child, travel, or moving to another country with higher risks.

#### 3.4.2. Parental Concerns, Perceptions, and Consequences of Choosing Not to Vaccinate

Reasons for not vaccinating include concerns about the number of vaccines given at once, age appropriateness of vaccines, and perceived unnecessary vaccines for certain diseases that are no longer prevalent.


*“[…] because I found it absolutely unreasonable to vaccinate such small children with such a load of vaccines, especially for example Hepatitis B, which to my knowledge is only transmitted through sexual intercourse and blood transfusions. I said, what does a 3-month-old baby have to do with such vaccination?”*

*(A07: 16)*


This statement captures some of the concerns and reasons mentioned for not vaccinating the children. The interviewee expressed concerns about the number of vaccines given at once, the age appropriateness of vaccines, and the perceived unnecessary vaccines for certain diseases (in this case, Hepatitis B) that might not be relevant for a 3-month-old baby.

Some parents believe that vaccines contain toxic substances that can harm their children, whereas others believe that the natural immune system is sufficient to protect them against diseases. Additionally, some parents do not trust the pharmaceutical industry and believe that the motivation behind vaccination is profit rather than health. Fear is also a recurring theme in parents’ reasons for not vaccinating, with some being more afraid of vaccine side effects than the diseases themselves.

The decision not to vaccinate has consequences, such as exclusion from social groups and schools, conflicts with family members, and fines or penalties from the government. Some parents have formed private kindergartens for unvaccinated children in response to their exclusion from public kindergartens.

#### 3.4.3. Motivation for Changes in Vaccination Attitude

Overall, there was little willingness to change vaccination attitudes. Many participants expressed a strong conviction that their current vaccination stance was right and that there was no alternative. However, some interviewees showed a hypothetical willingness to reconsider their vaccination attitude if the risk of not vaccinating increased, such as when traveling to a threatened area or when their children were older and had not yet had the disease.

Despite the low willingness to change vaccination attitudes, some participants showed ambivalence toward vaccination. They recognized that there were different opinions on the subject and that it was difficult to obtain a clear understanding of the issue. Ambivalence often arises within the context of a family background. For example, some participants struggled with conflicting information about the benefits and risks of vaccination, whereas others had partners or family members with different views on the subject. Ultimately, the decision to vaccinate is often personal or emotional.

This statement reflects the ambivalence toward vaccination and the challenges in finding clear information on the subject. Some participants expressed a low willingness to change their vaccination attitudes but recognized the existence of different opinions and found it difficult to form a clear understanding of the issue.

#### 3.4.4. Personal Beliefs, Risks, and Social Pressures in Decision Making

Generally, there is low willingness to change one’s vaccination stance. However, some interviewees hypothetically showed a willingness to reconsider their vaccination stance if the risks of not vaccinating increased, such as staying in a threatened area or if their children were older and had not yet contracted the disease. Ambivalence was also evident in the responses despite the low willingness to change their vaccination stance. This is particularly evident in relation to familial background. Findings on coping with compulsory vaccinations have revealed a range of avoidance strategies. There is no consistent pattern, but rather, a sense of ambivalence. This decision is often delayed, and some individuals avoid vaccination by deceiving or waiting until the last possible moment. Some attend meetings when invited, some pay fines, and others ignore all the requests.


*“Last year, we kind of cheated our way through.”*

*(A11: 44)*


The statement represents the avoidance strategies, ambivalence toward vaccination, delays in decision making, and low willingness to change their vaccination stance, but may reconsider if risks increase. It also highlights avoidance strategies, such as deception or waiting until the last moment, and how the decision-making process can be accompanied by doubts and feelings of guilt.

The findings of the reasons for vaccination reveal the circumstances under which the interviewees would vaccinate their children or themselves, whether real or hypothetical. One recurring reason is the individually higher perceived health risk of the disease compared with the corresponding vaccination, particularly in the case of polio, tetanus, and tropical diseases. The reduction in the number of mandatory vaccinations to a few, the possibility of individually choosing specific vaccinations, and less invasive administrations also feature as reasons that make the decision to vaccinate easier. Some are influenced by friends or their social environment. Another reason is the difficulty of reconciling families and working for mothers with unvaccinated children. Ultimately, deciding to vaccinate, even hypothetically, leads to doubts and feelings of guilt in vaccine-sceptical individuals.

#### 3.4.5. Factors and Personal Experiences Influencing COVID-19 Vaccination Decisions

The findings on factors influencing the decision to vaccinate against COVID-19 have limited significance because the criteria surrounding COVID-19 vaccination are constantly changing, and these interviews could not address the rapid developments in the last few months, from September to December 2021. One of the most important factors is the uncertainty and concerns about the immediate and long-term effects of the new vaccine. Some cited the personal experiences of people in their immediate circle who had suffered from stroke or other side effects after receiving the vaccine. There were also concerns about the unknown long-term effects of the vaccine and lack of information regarding the impact of the vaccine on the body. Many people feel that they would rather take the risk of getting infected with COVID-19 and dealing with its long-term effects than taking a vaccine with unpredictable side effects. The right to individual freedom of choice was also mentioned.

Regarding the reasons for not being vaccinated against COVID-19, the findings focused on the various reasons cited by people for not being vaccinated. These reasons are similar to those mentioned previously. Some people are sceptical of the pharmaceutical industry and its profit motives, whereas others are afraid of the potential negative health effects of the vaccine. Some people feel that they are healthy enough to withstand the virus without serious consequences, whereas others point to the experiences of family and friends who have survived the virus without being vaccinated. Some respondents expressed concern that vaccination would lead to conflicts and divisions within their families and friends. Some people actively seek alternative, non-vaccine-based ways of protecting themselves and are willing to make significant changes to their lifestyles to avoid vaccine-related restrictions.


*“I would not forgive myself, I think.”*

*(A07: 44)*


This statement reflects the concerns, uncertainties, and feelings of guilt associated with the decision to vaccinate or not vaccinate against COVID-19 based on personal experiences, fear of side effects, scepticism toward the pharmaceutical industry, and the desire for individual freedom of choice. This captures the emotional weight and personal responsibility of individuals when making vaccination decisions.

### 3.5. Social Impact Factors

This study revealed social factors influencing vaccination behaviour and identified 11 subcategories, including social environment, family background, biographical factors, and behaviour within the healthcare system.

#### 3.5.1. Environment and Family Background

Interviewees generally interacted in a heterogeneous environment, encountering both vaccine supporters and opponents. This heterogeneity is often reflected in both professional and personal spheres. The social environment plays a role in reinforcing pre-existing tendencies, as individuals are more likely to accept information that aligns with their beliefs and less likely to accept non-conforming views.

Family background also plays a significant role in vaccination behaviour. Most respondents came from families where vaccination was supported and followed according to the vaccination schedule. Discussions about vaccination within families occurred, but were mostly open and non-conflictual, and individuals were not necessarily influenced by their families’ opinions. The study also highlighted that the interviewed individuals’ parents belonged to a generation in which medical authority was more readily trusted and things were not questioned as much. Additionally, fewer vaccines were required during that time, which could have influenced the parents’ attitudes toward vaccination.


*“Well, yes and no, let’s say. Our environment with children and acquaintances also has a similar way of thinking. We do have this circle. However, I also had colleagues or friends, for example, who did not understand it at all. They said, ‘Why don’t you do it?’ I also said that everyone should make decisions for themselves.”*

*(A08: 16)*


This statement refers to interactions in a heterogeneous environment as a common experience among the interviewed individuals. The interviewees mentioned encountering vaccine supporters and opponents in their social circles, reflecting the diversity of their opinions. The importance of individual decision making in the context of vaccination is highlighted.

#### 3.5.2. Biographical Influence and Couple Relationship

Interviewees reported that their negative attitudes toward vaccinations typically developed over time and were not inherited from their parents. The higher vaccination acceptance among parents in the past could be attributed to personal beliefs or greater trust in the state and the healthcare system; however, the exact cause remains unclear.

Discussions regarding vaccinations within couples often begin after the birth of a child. Shared opinions were not always present from the beginning, and one parent may have been undecided or poorly informed about the topic. Ultimately, the opinions of parents who placed higher importance on vaccination tended to prevail. As for social impact factors, family, partners, friends, the healthcare system, medical professionals, the state, and media all play a role.


*“At the beginning it was difficult for him, not the vaccination, but otherwise going a different way with the children was a bit more difficult for him, swimming a bit against the current. However, by now, we have the same opinion in this case and that is fine. He is not afraid either when a child is sick or has even had a fever up to 40 °C, we had the confidence and always managed it ourselves.”*

*(A11: 34)*


This statement indicates that the interviewee’s partner initially had a different opinion on vaccination and child-rearing, which aligns with the shared opinions not always being present from the beginning. Discussions and understanding can develop over time, resulting in shared opinions on vaccination. The importance of trust and confidence in handling health-related decisions within families was highlighted.

#### 3.5.3. Healthcare System Behaviour

Interviewees perceived a one-sided dissemination of information, which left no room for doubt or counterargument. The information provided was considered insufficient, particularly regarding the vaccine’s ingredients and potential side effects. The behaviour of the healthcare system tends to lead to a reluctance to vaccinate. The interviewees felt that information would be lacking without actively seeking clarification. They also had the impression that a sceptical attitude and desire for information were met with rejection from healthcare personnel. The opinions of paediatricians were not highly valued, as they were not allowed to make independent decisions within the healthcare system.

A few interviewees discussed the vaccination letters they had received. From their statements, it is apparent that the letters were not remembered positively and the interviewees generally did not recall the exact content. The letters were perceived less as a source of information and more as a reminder of mandatory duty, which was not well-received. Therefore, the letters did not achieve their intended effects among interviewees.

The healthcare system has faced significant criticism for their handling of pandemics. Critiques were diverse and did not lead to clear conclusions. Some interviewees expressed that if the healthcare system had focused on advising people on how to strengthen their immune systems rather than instilling fear, the outcome might have been different. There was also uncertainty about the extent of decision-making power held by healthcare providers compared to politicians.

When asked about the behaviour of professionals, the interviewees reported their personal experiences in detail. They desire more comprehensive information, thorough conversations, and greater openness to alternative opinions and perspectives from professionals. There was little room for criticism and doubt about vaccinations and their ingredients in discussions with professionals. When interviewees expressed doubts about vaccinations, they felt that there was little room and described that their fears and concerns were not acknowledged. A critical perspective from professionals was also missing, and the interviewees felt that professionals did not critically evaluate vaccinations but instead unreflectively adopted others’ opinions.


*“Yes, because there is a very one-sided information: ‘Because vaccination only protects.’ Side effects are also hushed up; they are not acknowledged at all–that is, how it is. ”*

*(A15: 56)*


#### 3.5.4. State Regulations

Trust in the state regarding the rationale for introducing mandatory vaccinations was very low or nonexistent among the interviewees. They did not believe that the state prioritized citizens’ health or herd immunity. Other motives were suspected but were not discussed in detail. The imposition of mandatory vaccinations reinforced negative attitudes and divided society into those who followed the authority and those who resisted it. In general, no positive effects were attributed to mandate.

The interviewees were also sceptical of the state regulations to contain the pandemic and questioned the implemented measures. Some interviewees understood the intentions behind measures like the ‘green pass’ but felt that it was not the best solution from various perspectives. The ‘green pass’, also known as the COVID-19 Digital Green Certificate, is a digital or paper document introduced by the European Union during the COVID-19 pandemic to facilitate safe and unrestricted movement within the EU. The ‘green pass’ provides proof that a person has either (1) been vaccinated against COVID-19 with an approved vaccine, (2) recovered from COVID-19 and has natural immunity, or (3) tested negative for the virus through a recent PCR or antigen test. Others were puzzled by the changing regulations and how they were not well received in society, which they perceived as obedient to authority.


*“And what the state is doing here, it’s not about health.”*

*(A07: 82)*


There was low or no trust in the state among the interviewees, who suggested that the state’s actions were not focused on citizens’ health. The interviewees did not believe that the state prioritized citizens’ health or herd immunity. This statement emphasizes the scepticism toward state regulations to contain the pandemic.

### 3.6. Parenthood

The overarching category of parenthood (43 codes) describes the influence of being a parent of one or more children on attitudes toward vaccination and mandatory vaccination. The interviewed parents perceived a significant responsibility placed on them upon the birth of their children. They are accountable for the consequences of every decision they make, including mandatory vaccination. They believed that neither the state nor the healthcare system would assume this responsibility, even with the introduction of mandatory vaccinations. Therefore, they wished to make decisions regarding vaccination.

Parents’ desire to make the best decisions for their children is strong. Their statements revealed ambivalence, weighing the pros and cons of vaccination. Without claiming 100% certainty, they found the right decision was to refuse the mandatory vaccination. Interviewees believed that a standardized vaccination plan did not consider the individuality of each child. As parents, they feel that they know what their child needs and what they do not, which might not necessarily be the same for another child. Many parents decided to wait until their child was older before reconsidering possible vaccinations during the first few months of life.


*“I could not say now that it would have come from outside, but it started there, to have the feeling that I am now responsible for this child. ”*

*(A02: 2)*


The responsibility that the interviewed parents perceive upon the birth of their children is a central theme, where parents wish to make decisions regarding vaccinations themselves. This statement implies that responsibility comes from within and is not influenced by external factors, which aligns with the idea that parents believe that they know what their child needs.

### 3.7. Personal Impact Factors

The study’s focus on personal impact factors influencing vaccination behaviour was divided into eight subcategories: trust, experiences with illness and health, worldview, attitude toward illness, health medication, dealing with out-groups, self-image, and personality traits.

Trust in the healthcare system, government, and media has a significant impact on people’s decisions to vaccinate. Respondents reported that their trust in these institutions is weak or non-existent, and they provide various experiences that have led to this lack of trust.


*“I have to honestly say that I have increasingly less trust in the healthcare system, I just have to say it like that. I feel like it is about money and not health. If it were really about health, much more would have to be done in relation to alcohol, smoking, healthy nutrition, obesity, and so on. Many foods would have to be banned if you really wanted to ban, with sugar content or declarations would have to be much more precise about what is in all the finished products. If it were really about health.”*

*(A07: 22)*


This statement discusses concerns regarding the priorities of the healthcare system and the need for a more comprehensive approach to health, covering issues such as alcohol consumption, smoking, nutrition, and obesity. Respondents preferred natural healing methods, such as homeopathy or the body’s natural defences, over traditional medications.

Their worldview is strongly influenced by their belief in the power of nature, fate, and the natural order of things, with many respondents expressing scepticism about human intervention in these processes. They generally view childhood diseases as beneficial for strengthening the immune system and contributing to their overall development.

Lastly, respondents did not entirely reject conventional medicine, but tended to use alternative methods and self-inform before resorting to medication. They appreciate the benefits of modern medicine and trust in human biology. This study suggests that people’s vaccination behaviour is influenced by a combination of trust in institutions, personal experiences, and beliefs about health and natural processes.

### 3.8. Structural Impact Factors

This category encompasses four subcategories with a total of 150 codes: decision making freedom (74 codes), sanctions (40 codes), transparency (20 codes), and fear generation (16 codes). Two subcategories were specified for COVID-19: sanctions (three codes) and fear generation (six codes).

Many interviewees valued their freedom to make their own decisions, particularly regarding vaccinations for childhood diseases and COVID-19. They consider the ability to make independent decisions for their own health and that of their children to be fundamental rights in a democratic society. However, when faced with pressure and coercion, interviewees expressed feelings of unrest and viewed these measures as undemocratic and invasive. This can lead to a reactionary response in which the eliminated options become more important.


*“As I said, I am fundamentally in favour of free choice, and that’s why I told him, based on my intention, ‘If you feel better and protected afterwards, please do it. If you are sceptical, if you have any concerns, either clear them up or leave it be.’ I am really for absolute freedom.”*

*(A02: 50)*



*“I find coercion the worst thing there is because forcing a person, a healthy person, to undergo a physical intervention is simply not acceptable.”*

*(A15: 140)*



*“For me, this is not an open, mature society or a civil citizenship that can say, ‘I am here in a free country, and it is expected and trusted that I understand and make my own decisions’ […] I do not feel seen as a responsible citizen who has a mind to think for themselves, but rather I feel put down, like I am too stupid or too dumb to understand, or we do this now because otherwise, we will not get the vaccination rates at the national level.”*

*(A02: 38)*


The statements covered the themes of personal freedom in decision making, opposition to coercion, and the desire for an open, democratic society in which individuals have the autonomy to make their own choices regarding vaccination.

The findings also cover the sanctions interviewees faced owing to their vaccination decisions, primarily regarding mandatory childhood vaccination. Interviewees reported that their children or friends were excluded from kindergartens and received fines. These sanctions evoke feelings of anger and confusion. None of the interviewees viewed the sanctions as justified. The fines are seen as contradictory because paying them allows individuals to effectively “buy” their freedom from vaccination.

In the context of vaccination, transparency in reporting and information dissemination are critical structural factors that affect behaviour. Interviewees criticized the excessive focus on the benefits of vaccination, while risks and potential side effects were underreported. They expressed a desire for balanced reporting and information sharing, including acknowledgement of potential risks and challenges. Concerns have also been raised regarding data-privacy inconsistencies in vaccination procedures.


*“Of course, everyone is allowed to get vaccinated, that’s fine, but herd immunity and this obligation simply infringe on the human rights of every individual.”*

*(A11: 110)*



*“In any case, we had to pay the fine, it was about 200 €, and then we were left in peace.”*

*(A07: 16)*



*“These media outlets seem like a network that only speaks one truth, but everything else is silenced.”*

*(A11: 198)*


These statements represent themes of personal freedom, consequences of not vaccinating, and a desire for more balanced information sharing in the media. Fear generation is another structural factor that influences vaccination behaviour. This ethical question revolves around the appropriateness of using fear as a tool to persuade people to adopt specific behaviours, such as vaccination. Interviewees generally disapproved of the fear-based approach and compared it to authoritarian parenting. They argued that fear can make people more compliant but also lead to illness and poor decision making. Participants emphasized the importance of providing balanced information and respecting diverse opinions instead of relying on fear tactics to promote vaccination.

## 4. Discussion

This comprehensive study analysed the attitudes and beliefs of vaccine sceptics in Italy toward mandatory vaccinations and COVID-19 vaccinations. It identified two main categories of vaccination-related information: those related to mandatory vaccinations, which included subcategories such as information sources, knowledge, scepticism, and attitudes, and those related to COVID-19 vaccinations, which encompassed subcategories such as attitudes toward COVID-19, information sources, scepticism, and attitudes toward COVID-19 vaccinations. The results suggest that parents utilized a variety of information sources, including social networks, medical professionals, alternative medicine practitioners, books, websites, and media. The study identified five major themes contributing to vaccination scepticism: Hepatitis-B vaccination, the number of mandatory vaccinations in Italy compared to other countries, the economic interests of the pharmaceutical industry, knowledge of vaccine injuries, and the handling of mandatory vaccinations by legislation. Participants expressed concerns about mandatory Hepatitis-B vaccination and its administration timing, the necessity of mandatory vaccinations against certain diseases, the pharmaceutical industry’s motives, vaccine injuries, and the handling of mandatory vaccinations by legislation.

A key feature of Grounded Theory, which was applied for data analysis here, is the emphasis on the interconnection of data collection and analysis throughout the entire research process. Unlike traditional experimental research designs that require hypotheses to be formulated at the beginning and tested using a predetermined sample, Grounded Theory advocates the continuous generation of hypotheses based on the ongoing analysis of data. As new hypotheses emerge, the researcher adjusts the sample and research questions to strengthen, modify, or refute these hypotheses. Cyclical engagement with data, both during collection and analysis, is a defining characteristic of Grounded Theory. Hypotheses arise from working with the data, making the research approach data-driven, rather than hypothesis-driven. The continuous comparison between data, samples, and additional data to be collected aligns with the purpose of theory building, as described by Strauss and Corbin [20]. This process prioritizes the “discovery” of empirical findings in a specific field over theoretical preconceptions and leads to the study’s findings.

The emergent meta-category of “self-relatedness” plays a pivotal role in understanding participants’ perspectives and behaviours. This meta-category highlights the strong focus on individual and family well-being, as well as the interplay between personal experiences, social groups, and trust in the health system. The findings suggest that self-relatedness may contribute to vaccine scepticism and resistance to health campaigns, emphasizing the need for tailored communication that address the concerns and values of this population, while fostering trust and promoting informed decision making. “Self-relatedness” is a unique theme within the realm of vaccine hesitancy. This theme encapsulates how personal experiences and self-identity, often associated with holistic health practices and scepticism towards mainstream medicine, shape individual attitudes towards vaccines. It suggests that public health interventions aiming to address vaccine scepticism must consider these deeply personal and self-related factors, augmenting the traditional fact-based approaches. This theme echoes findings from previous research indicating the significance of personal beliefs and experiences in health decision-making processes [21]. The incorporation of such perspectives may aid in the design of more tailored and effective vaccine acceptance strategies.

Findings on attitudes toward COVID-19 vaccination revealed scepticism driven by concerns about vaccine development speed, novelty, and potential long-term damage [22]. Statements from medical professionals expressing doubt and suspicion about the profit motives of pharmaceutical companies and organizations, such as the WHO, further reinforced scepticism. In general, interviewees emphasized their doubts about the cost–benefit analysis concerning personal health, and possible side effects outweighed the perceived benefits of COVID-19 vaccinations for most people, particularly young individuals and children [23].

This study provides insight into the complexity of vaccine scepticism and highlights the need for better communication and education regarding vaccination. The findings confirm that vaccine scepticism is driven by a combination of factors, including distrust of the pharmaceutical industry and government, perceived inconsistency in vaccination policies, and concerns regarding the potential risks associated with vaccination [24]. The wider sociopolitical context in which vaccine hesitancy has grown, particularly in light of the COVID-19 pandemic, is acknowledged. Factors such as a perceived erosion of democratic governance and potential mistrust in pharmaceutical companies due to incidents like Pfizer’s USD 2.3 billion fine for deceptive promotion mechanisms, while not directly related to vaccines, can indirectly influence public perception of vaccines. Furthermore, questions about the rationale behind certain pandemic policies, such as lockdowns and mask mandates, have reportedly contributed to reduced public trust. This aligns with our study’s findings of mistrust as a significant factor in vaccine scepticism, suggesting that addressing trust issues on a broader level may be critical for improving vaccine acceptance.

In addition to exploring scepticism, this study aimed to identify the social factors that influence vaccination behaviour [25]. The findings confirm that attitudes toward vaccination are affected by various personal, relational, and structural factors [26]. The heterogeneous environment and family background appear to influence vaccination behaviour, with the social environment often reinforcing pre-existing beliefs. Interestingly, most respondents had family backgrounds that were supportive of vaccination, suggesting that external factors may play a more significant role in shaping negative attitudes.

Biographical influences and couple relationships also affect vaccination decisions, with attitudes typically evolving over time. Shared opinions about vaccination within couples often emerge after a child’s birth, with the parent placing greater importance on vaccination having a more significant influence [27]. Parenthood has emerged as an influential factor, with parents feeling a strong responsibility to make the best decisions for their children.

The healthcare system’s behaviour plays a crucial role in individuals’ attitudes toward vaccination, and the perception of one-sided information dissemination and lack of openness to alternative opinions or perspectives among professionals contribute to vaccination reluctance [28,29].

Low trust in state regulations and the perceived disconnect between the state’s priorities and citizens’ health also impact vaccination decisions [30,31]. The impact of trust has been widely confirmed for COVID-19 vaccination, including a representative survey performed in South Tyrol [10,32]. The introduction of mandatory vaccinations has polarized society, with some individuals becoming more resistant to vaccination due to the perceived imposition of their personal freedoms [33].

Personal impact factors, such as trust in institutions, experiences with illness and health, and worldviews also play a role in vaccination decisions. Structural impact factors, including decision making freedom, sanctions, transparency, and fear generation, also influence vaccination behaviour [34].

This qualitative study identified various social factors that influence vaccination behaviour. Addressing the complex interplay between personal, relational, and structural factors is essential to better understand vaccination decisions. Further research is needed to explore these factors more deeply and develop effective strategies to promote informed decision making and increase vaccination confidence. This study highlights the importance of understanding the multifaceted nature of vaccine scepticism and its various contributing factors.

For instance, healthcare professionals could engage in a more open dialogue with patients, present balanced information on vaccination benefits and risks, and address individual concerns. This approach can help foster trust and promote informed decision making [35]. Additionally, policymakers could consider alternative approaches to mandatory vaccination that respect individual autonomy while still promoting public health [36]. Public health campaigns could also focus on addressing distrust toward the pharmaceutical industry and government by emphasizing transparency and accountability in vaccine development, regulation, and implementation [37,38].

Education plays a vital role in attitudes toward vaccination. By developing comprehensive educational programs and providing accurate, evidence-based information on vaccines and their benefits and potential risks, it is possible to counteract misinformation and misunderstandings [39]. These programs should be tailored to different age groups, cultural backgrounds, and educational levels to ensure their maximum effectiveness.

Furthermore, it is essential to recognize the influence of social networks on vaccination decisions [40]. Public health campaigns can leverage the power of social networks and social media platforms to disseminate accurate information about vaccines, combat misinformation, and promote positive attitudes Engaging key influencers and opinion leaders within communities to promote accurate information about vaccines and address vaccine hesitancy could be an effective strategy [37]. However, influencers and opinion leaders may not necessarily be experts in the field of vaccines, immunology, or public health, which may lead to concerns about the accuracy and reliability of the information they share. They could create echo chambers, where only one narrative is promoted, limiting the diversity of opinions and potentially stifling healthy debate. The oversimplification of complex issues related to vaccines may lead to a misunderstanding of the risks and benefits of vaccination. As a strategy, such promotion of positive vaccine attitudes may be viewed as an attempt to manipulate personal beliefs and infringe on their right to make informed decisions about their health and the health of their families.

In conclusion, addressing the complex interplay of personal, relational, and structural factors that contribute to vaccine scepticism is essential for better understanding and potentially improving informed vaccination decisions. By acknowledging and addressing these factors, it is possible to develop effective strategies to promote informed decision making.

### 4.1. Action Strategies

Based on the study findings, the following strategies can be deduced to improve vaccination decision making:Improve communication and education: Address misinformation and misunderstandings by providing accurate, up-to-date, and comprehensive information on vaccinations through various channels including healthcare providers, media, and educational programs.Address distrust in the pharmaceutical industry and government: Encouraging transparency, promoting ethical practices, and demonstrating a genuine commitment to public health to rebuild trust in these institutions.Tailor vaccination policies: Consider the individual needs and concerns of different demographic groups when designing vaccination policies. This might include offering more flexible vaccination schedules or providing additional information on the rationale behind the mandatory vaccination.Engage healthcare professionals: Train healthcare providers to communicate effectively about vaccinations and address patient concerns. Encourage open dialogue and promote informed decision making.Foster trust in the healthcare system: Ensure that healthcare professionals provide balanced, unbiased information on vaccinations and remain open to alternative opinions and perspectives.Address concerns about personal freedoms: Consider alternative approaches to promoting vaccination that respect individual autonomy and choice.Support parents in making informed decisions: Provide resources and guidance to help parents make well-informed decisions about their children’s vaccinations, considering their unique circumstances and needs.Avoid fear tactics and coercion: Focus on promoting vaccination through education, encouragement, and positive reinforcement rather than using fear or punitive measures.Conduct further research: Invest additional research to explore the social factors influencing vaccination behaviour in more depth and to develop effective strategies that promote informed decision making.

### 4.2. Limitations

This qualitative study has several limitations. (i) The study was based on a small sample of ten participants, which may not be representative of the broader population of vaccine-sceptical parents in South Tyrol or elsewhere. However, these findings may not be generalizable to other contexts or populations. (ii) For participant selection, the study applied purposive sampling to ensure the inclusion of parents holding sceptical attitudes towards vaccination, as determined by preliminary data from an ongoing quantitative study. This approach aimed to capture a breadth of perspectives on vaccine hesitancy. Nevertheless, the recruitment faced challenges due to observed ambivalence and withdrawal after initial unofficial commitment, limiting the final pool of participants to ten. It is important to note that the trust and willingness to participate among such a sceptical population present inherent difficulties, and although the sample size is small, it provided valuable insights into this public health concern. The purposive sampling method used to select participants might have introduced a selection bias, as the researcher intentionally chose participants based on their vaccine-sceptical attitudes. This approach could limit the diversity of perspectives and experiences captured in the study. (iii) Qualitative research is inherently subjective, as it relies on the researcher’s interpretation of participants’ responses. This subjectivity could introduce potential bias and affect the reliability of the findings. The study did not provide quantitative data to support the findings or measure the extent of the observed attitudes and behaviours. This study acknowledges potential limitations due to the utilization of three interviewers. Originally trained to manage a larger sample, the three interviewers carried out interviews with a smaller than expected participant group due to recruitment challenges. This may have introduced some variations in data collection, as each interviewer might have influenced responses differently. However, our rigorous training and briefing for each interviewer aimed to minimize such bias, striving for consistency across all interviews. (iv) Participants might have provided responses that they believed the researcher wanted to hear or that they would present themselves in a positive light, which could affect the validity of the study. (v) The study focused on vaccine-sceptical parents in South Tyrol, potentially overlooking other relevant stakeholders, such as healthcare providers, educators, and policymakers, who could provide additional insights into vaccination hesitancy and refusal. (vi) This study captures participants’ attitudes and opinions in 2021, which may not reflect how their views evolve over time in response to new information or changing circumstances.

## 5. Conclusions

This study presents an in-depth analysis of vaccine scepticism and attitudes toward mandatory and COVID-19 vaccination in Italy. This highlights the complexity and multifaceted nature of the factors that influence vaccination behaviour, including personal experiences, family background, biographical influences, healthcare system interactions, and trust in institutions. This study emphasizes the need for improved communication and education to address the concerns and misconceptions surrounding vaccination. Additionally, it underlines the importance of considering the diverse factors that shape individuals’ attitudes when developing strategies to promote informed decision making.

The findings of this qualitative study contribute significantly to our understanding of the social factors driving vaccination scepticism and behaviour. It is evident that a one-size-fits-all approach to vaccination may not be effective, given the myriad personal, relational, and structural factors at play. Therefore, future research should focus on exploring these factors in greater depth to address vaccine hesitancy.

## Data Availability

The datasets used and/or analysed during the current study are available from the corresponding author upon reasonable request.

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
