# Peer review of "Unravelling Vaccine Scepticism in South Tyrol, Italy: A Qualitative Analysis of Personal, Relational, and Structural Factors Influencing Vaccination Decisions"

_healthcare, 2023, doi:10.3390/healthcare11131908_

Round 1

Reviewer 1 Report

This a qualitative study attempting to understand vaccine skepticism in northern Italy. This is certainly an important public health issue. My comments listed below focus only on the concerns.

1. The study rationale in the introduction is weak. The authors argue that there is an association between "social vulnerability" and vaccine hesitancy, but they do not elaborate the concept of "social vulnerability" and why.

2. The justification of using qualitative methods to study the vaccine hesitancy is not strong. The authors should explain why qualitative methods are appropriate in answering the research questions.

3. The description in the methods section is not clear. For example, in line 150, "interviews were conducted by three individuals." So there were 3 interviewers. in line 151 "All participants..." refers to the 3 individuals (they were actually the interviewers) or the interviewees? 

4. Why 3 interviewers were needed to interview for interviewing just 10 study participants? In qualitative studies, the interviewer per se is the study instrument. Three different interviewers means three instruments were used in the data collection. Potential biases might be introduced in the data collection process.

5. It is true that sample size is less an issue in a qualitative study. However, the inclusion of study participants should pay attention to whether all key informants (in terms of the knowledge of and perspectives on the topic) have been included? The authors might want to explain why and how these 10 study participants were selected.

6. Finally, the interpretation of the data is not well-presented. The authors should help readers understand how the interpretations were derived from the data (the interviews).

Reviewer 2 Report

I highly appreciate the choice to conduct research on vaccine hesitancy in a rural region that exhibits hesitancy, as opposed to the more commonly used approach of studying easily accessible medical students. However, it is important to note that the sample size of 10 in the study is a very serious limitation that cannot be rectified at this point. One question that arises is whether such a small number of individuals with higher education is representative of the broader population in that region, or if the sample is skewed with that respect.

To provide a deeper understanding, it would be valuable to place these findings within the wider context of vaccine hesitancy studies. By examining other research, you can demonstrate that the objections raised by the study participants are not unique to this region but are often voiced by a significant portion of the population. This highlights the importance of acknowledging and addressing these concerns, especially among those who have been not noticed as were unenthusiasticly following regulations or those who, after losing trust in established institutions, have sought out information on their own and come to the conclusion that the provided vaccines do indeed seem to be effective.

For example, there have been numerous studies conducted during the pandemic in several seemingly successful countries, which have indicated a significant proportion of the population expressing the opinion that they no longer perceive their country as being governed democratically. This sentiment awkwardly aligns, to some extent, with subsequent studies referring to what is known as the "erosion of democracy." It is worth noting that the suggestion that pharmaceutical companies may prioritize profit motives over public welfare is not without concern. This is exemplified by Pfizer's $2.3 billion fine for engaging in deceptive promotion mechanisms, including making misleading claims about the safety and efficacy of their drugs. Although these charges did not pertain to vaccines specifically, the record-breaking fine raises justified questions about the level of trust that can be placed in at least that particular company. Additionally, recent studies indicate that the rationale behind many pandemic policies, such as lockdowns and mask mandates, remains somewhat uncertain. It would be beneficial to consider studies on trust as well, as the lack of trust was one of the key challenges during the pandemic, and many of the adopted policies may have inadvertently contributed to a further reduction in public trust.

Reviewer 3 Report

Dear colleague,

Thank you for the kind invitation to review the above manuscript.

Attached are my comments for the authors' consideration

Use mesh terms for keywords

Introduction

- to include other qualitative studies performed in Italy as well in the international platform

- What is unique about South tyrol that would warrant an evaluation of covid-19 scepticism in the population

-> Will be helpful to provide some basic information about the unique demographics of the population

Methods

- To attached the COREQ checklist

Discussion

- Are there any unique findings / themes identified in the study

-

Minor comments

High rates of covid-19 hesitancy globally should be briefly highlighted 

-> https://pubmed.ncbi.nlm.nih.gov/34452026/

Minor grammatical errors 

Round 2

Reviewer 1 Report

The revised manuscript has addressed my concerns. I have no more questions.